# Characterization of the Effects of Host p53 and Fos on Gallid Alpha Herpesvirus 1 Replication

**DOI:** 10.3390/genes14081615

**Published:** 2023-08-12

**Authors:** Zheyi Liu, Lu Cui, Xuefeng Li, Li Xu, Yu Zhang, Zongxi Han, Shengwang Liu, Hai Li

**Affiliations:** 1Division of Avian Infectious Diseases, State Key Laboratory for Animal Disease Control and Prevention, National Poultry Laboratory Animal Resource Center, Harbin Veterinary Research Institute, The Chinese Academy of Agricultural Sciences, Harbin 150069, China; 2Department of Pathogenic Microbiology and Immunology, School of Basic Medical Sciences, Xi’an Jiaotong University, Xi’an 710061, China

**Keywords:** alphaherpesviruses, p53, Fos, virus–host interactions, virus replication

## Abstract

Treatment options for herpesvirus infections that target the interactions between the virus and the host have been identified as promising. Our previous studies have shown that transcription factors p53 and Fos are essential host determinants of gallid alpha herpesvirus 1 (ILTV) infection. The impact of p53 and Fos on ILTV replication has ‘not been fully understood yet. Using the sole ILTV-permissive chicken cell line LMH as a model, we examined the effects of hosts p53 and Fos on all phases of ILTV replication, including viral gene transcription, viral genome replication, and infectious virion generation. We achieved this by manipulating the expression of p53 and Fos in LMH cells. Our results demonstrate that the overexpression of either p53 or Fos can promote viral gene transcription at all stages of the temporal cascade of ILTV gene expression, viral genome replication, and infectious virion production, as assessed through absolute quantitative real-time PCR, ILTV-specific RT-qPCR assays, and TCID_50_ assays. These findings are consistent with our previous analyses of the effects of Fos and p53 knockdowns on virus production and also suggest that both p53 and Fos may be dispensable for ILTV replication. Based on the synergistic effect of regulating ILTV, we further found that there is an interaction between p53 and Fos. Interestingly, we found that p53 also has targeted sites upstream of *ICP4*, and these sites are very close to the Fos sites. In conclusion, our research offers an in-depth understanding of how hosts p53 and Fos affect ILTV replication. Understanding the processes by which p53 and Fos regulate ILTV infection will be improved by this knowledge, potentially paving the way for the development of novel therapeutics targeting virus–host interactions as a means of treating herpesvirus infections.

## 1. Introduction

The avian infectious laryngotracheitis virus (ILTV), also known as gallid alphaherpesvirus 1 (GaHV-1), is a member of the Herpesviridae family and the subfamily Alphaherpes-virinae. It manifests as an acute, contact-transmitted illness of the chicken upper respiratory tract [1]. With the rapid rate of transmission, the infection rate of ILTV is as high as 95–100%. Because of the characteristic that it cannot be cleared by any current means after infection [1,2,3,4], it emerges as one of the most significant infectious illnesses imperiling China’s chicken sector. Currently, ILTV is mainly prevented and controlled by the administration of a live attenuated vaccine, which plays an important role in China and even around the world. However, due to the development of a latent infection in the trigeminal ganglia following an acute upper respiratory tract infection, the generally used attenuated ILTV vaccine strain could undergo gene recombination in immunized chickens to form a new strain with stronger virulence [5,6,7,8,9,10,11,12]. Creating novel treatment approaches that concentrate on the interactions between ILTV and host cells, beyond the regulation of the host immune system, is an alternate method for controlling ILTV. Such therapeutics could potentially provide a more effective means of preventing and controlling ILTV infections.

p53, which has been named for its 53KD mobility in SDS-PAGE gel, is widely recognized as a tumor suppressor. It can be activated by many known stress signals in the body, and the biological effects caused by its activation could be very different among individuals, tissues, and even different types of cells [13]. In recent years, many new regulatory functions of p53 have been progressively discovered. Among these functions, p53 has been confirmed to regulate the infections of a sea of viruses [14], including vesicular stomatitis virus [15], poliovirus [16], and influenza A virus [17]. Marek’s disease virus, in particular in chickens, has to engage directly with chicken p53 (chp53) and its Meq oncoprotein in order to block the transcriptional activity and apoptosis that is mediated by chp53 [18]. The efficient inhibition of the replication of ALV-J [19] and infectious bursal disease virus [20] by chp53 has been also reported. Fos is a member of the proto-oncogene Activator protein 1 (AP-1) family. Members of this family are dimer transcription factors involved in the regulation of a variety of cellular processes including proliferation, apoptosis, differentiation, survival, migration, and transformation. In our previous studies, both chicken p53 and Fos have been found to be important factors in the infection of ILTV, including replication and transmission, since knockdown of either p53 or Fos could reduce the titer of ILTV and copies of the viral genome significantly [14,21,22]. However, the effect on viruses after an overexpression of p53 and Fos has not been clarified in detail yet. In particular, both p53 knockdown and Fos knockdown lead to the repression of the transcription of the virus’ immediate early gene *ICP4*. AP-1 family members act as cofactors of p53 in its transcriptional regulation of many genes, like matrix metalloproteinase-1 [23]. Our previous study has identified *ICP4* as the direct target gene of host Fos [14]. Whether p53 controls the transcription of *ICP4* by cooperating with Fos has not been elucidated, which is important for further illustration of the mechanisms of which regulate ILTV infection.

The purpose of this work was to examine the effects of overexpressing chicken p53 and Fos in LMH cells on each step of ILTV replication, including viral gene transcription, viral genome replication, and infectious virion generation. This recent study, together with our previous investigations, provide a detailed understanding of the roles played by p53 and Fos in ILTV infection. Further, using CoIP and ChIP-qPCR biological techniques, the potential cooperation between Fos and p53 during their control of ILTV infection was also investigated.

## 2. Material and Methods

### 2.1. Viral Strain and Cell Culture

At the Harbin Veterinary Research Institute of CAAS, a kind of virulent strain called ILTV-LJS09, which has been registered in the GenBank Accession by the number of JX458822, is kept in storage. Leghorn male hepatoma (LMH) cell line (ATCC CRL-2117) that has been chemically immortalized may be used to replicate this strain, and obvious CPEs can be seen. The cells were kept alive in DMEM (Dulbecco’s Modified Eagle’s Medium), which also contained 10% fetal bovine serum (FBS), 100 units/mL of penicillin, 100 g/mL of streptomycin, and 2 mM L-glutamine. Cell culture incubations were performed at 37 °C and 5% CO_2_.

### 2.2. Plasmids and Transfection

The expression vectors pFLAG-pCAG were subcloned with chicken p53 cDNA to produce the recombinant plasmid pCAG-p53-Flag. In order to create the recombinant plasmid pCAG-Fos-HA, chicken Fos cDNA was subcloned into pHA-pCAG as well. Twelve hours prior to transfection, the LMH cells were seeded onto tissue culture plates. We added 1 μg plasmid to each cell hole of the 24-well plate as recommended by the reagent manual, and transfection was conducted by the PEI transfection reagent (R0531, Thermo Scientific, Rockford, IL, USA).

### 2.3. RNA Extraction and RT–qPCR 

24-well plates were inoculated with LMHs. ILTV was introduced into the cells at a certain MOI (multiplicity of infection) of 1. Utilizing RNAiso Plus reagent from Takara Biotechnology, Dalian, China, total RNA was extracted from the virus-infected cells. The 24-well plate’s growth media was removed, and the cells underwent two PBS (Phosphate-Buffered Saline) washes. Each well received an addition of RNAiso Plus reagent before the fluid was transferred to 1.5 mL Eppendorf tubes. Each Eppendorf tube received trichloromethane to facilitate the extraction of RNA. The mixture was completely mixed, and the RNA was separated into the upper aqueous phase by centrifuging it at 12,000× *g* for 15 min at 4 °C. The aqueous phase’s top layer was carefully transferred to fresh Axygen tubes, and isopropyl alcohol in an equivalent amount was added to precipitate the RNA. The RNA was precipitated at 4 °C for 15 min, and then the RNA pellet was collected by centrifugation. The extra fluid was dumped. Then, 75% ethanol was used to wash the RNA pellet. The cleaned RNA was then dissolved in DEPC water. A specific primer design software, which is called Oligo 7, was used to design primer sequences for different genes. Using a One Step TB Green^®^ PrimeScriptTM RT-PCR Kit II (Takara Biotechnology, Dalian, China), real-time quantitative PCR (RT-qPCR) was carried out. Each sample was measured three times. The 2-Ct technique was used to quantify the relative quantification of target gene expression, and the findings are shown as the log_2_ fold change.

### 2.4. Preparation of Standards for Absolute qRT-PCR

Using certain primers, the PCR products of the ILTV genes ICP4, ICP27, gC, gI, and gG were amplified. Standard procedures were followed for conducting the PCR reactions. Using a special cloning kit, the amplified PCR products were cloned into the pMD18-T plasmid vector (Takara, Shiga, Japan). The aforementioned genes have been given PCR amplification primers (Table 1). The cloning procedure was adopted strictly by the kit manufacturer’s methods. The recombinant plasmids were converted into competent Escherichia coli DH5 cells after being ligated (Takara, Shiga, Japan). The transformed cells were then plated onto kanamycin antibiotic-containing selective agar plates to look for successful transformants. Plasmid DNA was recovered from the bacterial cultures using a plasmid extraction kit or a comparable technique utilizing the positive transformants that were chosen. The cloned PCR products were present in this DNA. To create standard curves for absolute measurement, the plasmid DNA was diluted to a series of known quantities. To precisely measure the target genes in the qRT-PCR reactions, the dilution series often encompassed a range of concentrations.

### 2.5. Absolute Quantitative Real-Time PCR

Using Luna Universal qPCR Master Mix (NEB, Ipswich, 3003L, MA, USA), the absolute qRT-PCR reaction mix was carried out in accordance with the manufacturer’s instructions. We separated the PCR reaction mixture into separate PCR plates. Then, we started the amplification by placing them in the real-time PCR device. Denaturation, annealing, and extension phases were often included in the cycle conditions. With three technical duplicates for each response, at least three separate experiments were conducted. 

### 2.6. Protein Extraction

To eliminate any outside pollutants, cells were rinsed with ice-cold phosphate-buffered saline (PBS). The cells’ soluble proteins were removed using a cell lysis buffer. Tris-HCl 100 mM (pH = 8), 150 mM NaCl, 1% NP-40, and phosphatase/protease inhibitor cocktail tablets from Abcam (Shanghai, China) were used in the lysis solution. The manufacturer’s procedure was followed during the extraction. Using a BCA Kit from Beyotime (China), the protein concentration in each sample was calculated. The BCA technique was used in this kit to determine the protein concentration.

### 2.7. Co-Immunoprecipitation Assay

Proteins were isolated from 2 × 10 cm plates of fully confluent LMH cells and utilized in each immunoprecipitation experiment (IP) in accordance with the manufacturer’s instructions (Invent Biotechnologies, Minnesota, MN, USA). The protein lysates were first added to 20 L Protein A/G Magnetic Beads (Santa Cruz Biotechnology, Santa Cruz, CA, USA) that had been previously cleaned with PBST, and rotated for 2 h at 4 °C with specific antibodies targeting against HA (Beyotime, China), Flag (Beyotime, China), or IgG negative control (CST, Danvers, MA, USA). One-tenth of the total protein extraction volume is used as input. The immunoprecipitation complexes were then isolated from the magnetic beads and rinsed four times with PBST buffer before being subjected to further Western blot analysis.

### 2.8. Western Blot

Western blotting was carried out completely in accordance with the methods previously described [24]. Electrophoresis on a polyacrylamide gel with sodium dodecyl sulfate (SDS-PAGE) was used to separate an equal quantity of protein from each sample. In accordance with their corresponding concentrations, the proteins were denatured and placed onto the gel. Using an electroblotting device, the separated proteins were transferred from the SDS-PAGE gel onto a nitrocellulose membrane (Yamei, China). For subsequent investigation, this procedure aids in immobilizing the proteins onto the membrane. 5% non-fat milk was used to block the transplanted membrane for two hours at room temperature. Blocking lowers background noise and aids in preventing nonspecific antibody binding. After that, primary antibodies directed against the relevant proteins were incubated on the membrane for the whole night at 4 °C. Antibodies that recognize HA, Flag, and tubulin were applied in this instance. As previously indicated, the antibodies were purchased from Proteintech in Wuhan, China, and Beyotime in China. The membrane was treated with a secondary antibody coupled to an enzyme or a fluorescent dye after the excess primary antibodies were washed off. The main antibody is recognized by the secondary antibody, which then attaches to it and increases the detection of the target protein. Using the proper detection techniques, such as chemiluminescence or fluorescence, the presence of the target proteins was seen. This procedure aids in identifying protein bands on the membrane.

### 2.9. Viral Quantitation 

LMHs were infected with the ILTV at MOIs of 0.01 or 1 according to the specific experimental procedures. The indicated MOI was obtained according to the number of cells to be infected and the estimated number of infectious particles, based on plaque-forming units detected in LMH cells. The levels of virus replication were determined using ILTV-specific RT–qPCR assays and TCID_50_ assays, as previously described [14]. To determine the total level of viral replication, both cell-associated viruses and the viruses released into the supernatant were collected for virus quantification. Cells were lysed via three rounds of freezing–thawing. As per the specified experimental protocols, LMHs were exposed to the ILTV at MOIs of 0.01 or 1. Based on plaque-forming units found in LMH cells, the specified MOI was calculated based on the expected number of infected cells and the projected number of infectious particles. Utilizing the previously described 14 TCID50 and ILTV-specific RT-qPCR assays, the amounts of viral replication were determined. Cell-associated viruses and viruses discharged into the supernatant were both collected for virus quantification in order to gauge the overall amount of viral replication. Through three cycles of freezing and thawing, cells were lysed.

### 2.10. Chromatin Immunoprecipitation (ChIP) Assays

LMH cells were infected with ILTV 12 h after being transfected with pCAG-p53-Flag or pCAG-Fos-HA for 24 h by PEI. After that, formaldehyde was used to crosslink the cells so that the protein–DNA connections could be preserved before the ChIP experiments could begin. We directly added 1% formaldehyde to the cell culture medium, and then let it sit there for 10 min at room temperature. Then, we added glycine to a final concentration of 125 mM and incubated for 5 min at room temperature to terminate the crosslinking reaction. We scraped the cells from the culture dish after gently washing them in cold phosphate-buffered saline (PBS). Then, the cell suspension was centrifuged at 1000× *g* for 5 min at 4 °C and the cell pellet was redissolved in lysis solution that contains a protease inhibitor cocktail and incubated for 10 min on ice. Then, we used a sonicator to sever the DNA in the lysate into fragments that are typically 200–500 base pairs in size. Each ChIP experiment used 5 g of either anti-HA (Sigma-Aldrich, St. Louis, MO, USA, mouse, # H9658) or isotype control IgG1 (Cell Signaling, Beverly, MA, USA, mouse, 5415) antibodies to shear chromatin samples from LMH cells (5 × 10^6^ cells). The manufacturer’s recommendations were followed while using Protein A/G PLUS-agarose beads, which is a kind of agarose coagulate. (Santa Cruz Biotechnology, Santa Cruz, CA, USA). The PCR Purification Kit, which was purchased from QIAGEN, Valencia, CA, USA, was used to clean the immunoprecipitated DNA. qRT-PCR was used with Bio-Rad CFX96 equipment to identify the ICP4 promoter region using primers unique to the promoter DNA. Three duplicates of each response were carried out.

### 2.11. Statistical Analysis 

All statistical analyses were performed using the GraphPad software suite (GraphPad Prism for Windows version 8.0, SPSS, San Diego, CA, USA, www.graphpad.com accessed on 4 March 2022). The mean standard ± deviation (SD) of the data gathered from several experiments are presented. The significance of differences between two groups was determined with two-tailed Student’s *t* test. For all analyses, * *p* < 0.05 indicates the levels of significance.

## 3. Results

### 3.1. Manipulating the Expression of p53 and Fos in LMH Cells

In order to verify the biological effects of the activations of chicken p53 and Fos on ILTV replication, an overexpression of p53 or Fos was performed in LMH cells by transferring pCAG-p53-Flag and pCAG-FOS-HA plasmids into LMH cells. An empty pCAG vector was used as a negative control. The transcription of chicken *Tp53* and *Fos* was detected by RT-qPCR. Significant inductions of the mRNA levels of both the key host factors were observed compared with those of the control group (Figure 1A,B). The overexpression of p53 and Fos proteins was validated by Western blotting using antibodies targeting HA and Flag, respectively (Figure 1C,D). The successful inductions of p53 and Fos proteins were further evidenced by indirect immunofluorescence by detecting the expression of HA and Flag (Figure 1E,F). Thus, the transfections of pCAG-p53-Flag and pCAG-FOS-HA plasmids induced the overexpression of p53 and Fos successfully in LMH cells. 

### 3.2. Overexpression of p53 Promotes Viral Gene Transcription

To gain insight into the effects of p53 overexpression on viral gene transcription, the mRNA levels of four ILTV genes covering all stages of the viral gene expression of ILTV, including the immediate early gene (IEG) *ICP4*, early gene (EG) *ICP27*, early/late gene (E/LG) *gI*, and late gene (LG) *gG* [13], were detected by absolute quantitative real-time PCR (absolute qRT–PCR) at 6 h post infection in LMH cells upon the overexpression of p53. The mRNA levels of all viral genes detected were significantly increased by p53 overexpression (Figure 2). 

### 3.3. Overexpression of p53 Promotes the Replication of ILTV

To explore the effects of p53 overexpression on viral replication in LMH cells, ILTV-specific RT–qPCR and TCID_50_ assays were performed to detect viral genome replication and the production of infectious virions, respectively. The level of the viral genome was increased significantly by pCAG-p53-Flag transfection at 48 h post infection (Figure 3A), together with a 13.8 times increase in the production of infectious virions (Figure 3B).

### 3.4. Overexpression of Fos Promotes Viral Gene Transcription

To assess the impact of Fos overexpression on the replication of ILTV genes, absolute qRT–PCR was conducted to measure the mRNA levels of the four ILTV genes above mentioned at 6 h post-infection in LMH cells with overexpressed Fos. The mRNA levels of all detected viral genes were significantly increased upon Fos overexpression (Figure 4). 

### 3.5. Overexpression of Fos Promotes the Replication of ILTV

To explore the effects of Fos overexpression on viral replication in LMH cells, ILTV-specific RT–qPCR and TCID_50_ assays were performed to detect viral genome replication and the production of infectious virions, respectively. The level of the viral genome was increased significantly by pCAG-Fos-HA transfection at 24 and 48 h post infection (Figure 5A), together with a 17.80 times increase in the production of infectious virions (Figure 5B).

### 3.6. Direct Physical Interaction between p53 and Fos Proteins

AP-1 family members are well-known cofactors of p53 in mammals. To determine if there is any direct interaction between chicken p53 and Fos proteins in our model, CoIP experiments were conducted. In LMH cells co-transfected with pCAG-Fos-HA and pCAGGS-p53-Flag plasmid, multiple binds of overexpressed Fos ranging from 40 kDa to 60 kDa were observed in IP-HA samples, and only the short bind was the main one bound by p53 (Figure 6A). The bindings of p53 binds (ranging from 41 kDa to 55 kDa) to Fos were also observed (Figure 6B).

### 3.7. ICP4 Is a Bona Fide Target Gene of Both p53 and Fos

AP-1 family members have been shown to be important cofactors of p53 in regulating the transcription of p53 target genes. Our previous study has identified *ICP4* as the direct target gene for Fos [21]. Considering the direct physical interaction between p53 and Fos proteins (Figure 6), we next addressed the question of whether p53 could also locate Fos-binding elements in the promoter region of *ICP4*. Inconsistently with our previous findings, four Fos binding sites, Fos-1 (F-1), Fos-2 (F-2), Fos-3 (F-3), and Fos-4 (F-4), were ensured by chromatin immunoprecipitation followed by qPCR (ChIP-qPCR) in LMHs (Figure 7A,B). The same primers for detecting these four Fos binding sites were used to detect p53-bound DNA in LMH cells upon the ectopic expression of p53. A negative control was employed by DNA input, and the isotype IgG1 antibody was tested through ChIP to perform an unspecific binding level between the DNA and antibody. Interestingly, all of these four sites were also bound by p53 (Figure 7C), suggesting the co-binding of p53 and Fos to the same sites in the promoter region of *ICP4*. 

## 4. Discussion

Our previous studies observed a significant repression of ILTV replication upon the knockdown of either p53 or Fos in LMH cells [21,22]. Detailed characterization of the effects of p53 and Fos on ILTV replication is required for further elucidation of the mechanisms by which p53 and Fos control ILTV infection, which remains unclear. The present study addressed this issue by manipulating the expression of p53 and Fos in the ILTV permissive chicken cell line. We found that both p53 and Fos have promoting effects on each stage of ILTV replication.

The expression of genes during herpesvirus infection occurs sequentially, with the protein encoded by the first expressed gene regulating the expression of other genes. Most of the products of immediate early genes are transcription factors whose expression triggers the initiation of cascade regulation. Meanwhile, early genes encode the enzymes involved in nucleotide metabolism and DNA replication, as well as some envelope glycoproteins, and late genes typically encode structural proteins and other proteins involved in virion assembly [13]. Despite the repression of the transcription of ILTV IEG and EG, virus–host metabolism interactions are promising targets for developing therapeutics against herpesviruses. Our previous metabolomics study explored the metabolic requirement of ILTV replication, among which some key host enzymes regulating nucleotide metabolism and ATP synthesis have been found under the regulation of both p53 and Fos [21,22,25]. Therefore, the co-regulation of metabolic processes by p53 and Fos may contribute to the control of ILTV replication that we observed in the present study and is worthy to be studied further. 

Previous studies have revealed the significance of studying the transcriptional regulation of *ICP4*, an immediate early gene of ILTV. The transcriptional activity of *ICP4* is crucial for initiating gene transcription in other stages of the virus. One of the findings from earlier research was that Fos can target the upstream region of *ICP4*, leading to an increase in its transcriptional activity [21]. Therefore, it is particularly important to study the transcriptional regulation of *ICP4*, which might provide novel therapeutic strategies by preventing the transcriptional cascade of viral gene transcription from the beginning. Here, we found that there was a direct physical interaction between p53 and Fos proteins, and both p53 and Fos bind to the promoter of *ICP4* with shared binding elements. Previous studies have revealed that the binding of p53 per se is not sufficient to regulate gene expression; it probably plays a biological function via the recruitment of additional factors [26]. It is yet to be determined whether both p53 and Fos directly target the upstream of *ICP4*, or if one targets the upstream of *ICP4* depending on the other. Furthermore, p53 also plays an important role in infection in other herpesviruses, for example, it has been illustrated that p53 can regulate the transcription of the immediate early genes *ICP27* and *ICP0* of HSV-1 [27]. Hence, understanding the mechanism and implications of the interaction between p53 and Fos in the transcriptional regulation of *ICP4* is crucial and requires further exploration.

In conclusion, our study provides a detailed characterization of the effects of host p53 and Fos on gallid alpha herpesvirus 1 replication, which extends our understanding of ILTV host interactions. These findings can be valuable for future studies aiming to elucidate the mechanisms by which p53 and Fos control ILTV infection. By identifying the role of these transcription factors in ILTV replication, we may gain insight into potential targets for the development of antiviral therapies. Overall, this study contributes to our understanding of the molecular mechanisms underlying herpesvirus infections and may lead to the development of new strategies for controlling these infections.

## Figures and Tables

**Figure 1 genes-14-01615-f001:**
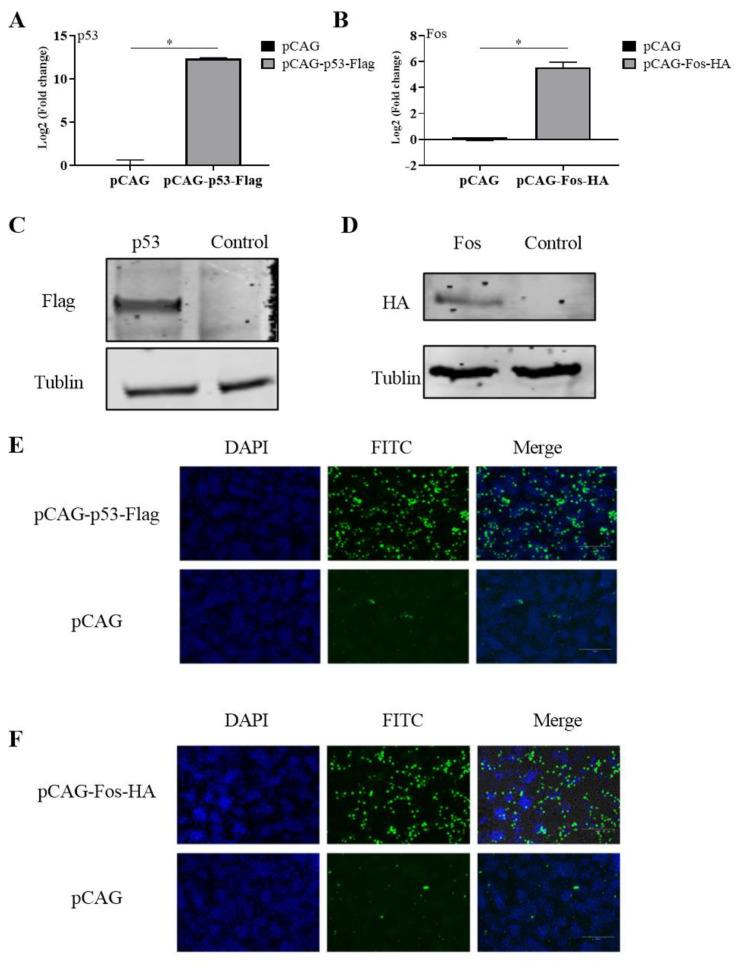
Manipulating the expression of p53 and Fos in LMH cells. For the overexpression of chicken p53 or Fos, LMH cells were transiently transfected with pCAG-Fos-HA or pCAG-p53-Flag for 24 h. (**A**,**B**) The transcription of *Tp53* or *Fos* was analyzed by RT-qPCR. (**C**,**D**) The induction of p53 or Fos protein was detected by Western blotting or (**E**,**F**) indirect immunofluorescence using antibodies targeting HA or Flag, respectively. Empty pCAG was used as negative control in all overexpression experiments. The scale bar indicates 150 µm. Tublin was used as inner control in Western blotting. Data are presented as the mean ± SD, *n* = 3. * *p* < 0.05 indicates the levels of significance.

**Figure 2 genes-14-01615-f002:**
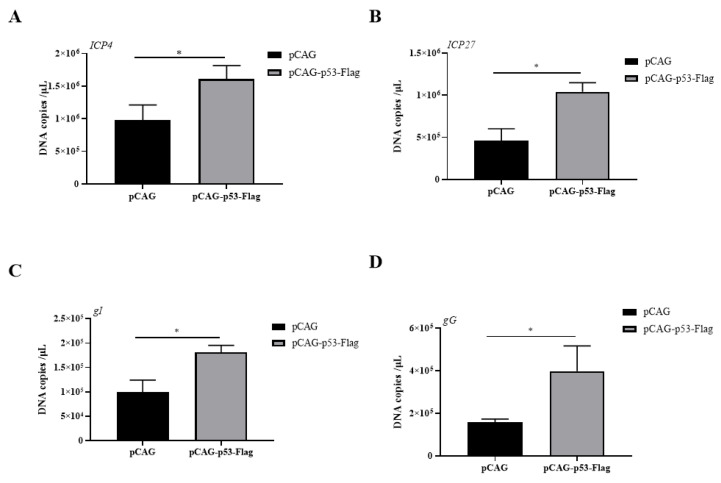
Detection of viral gene transcription upon p53 overexpression in LMH cells. The mRNA levels of four ILTV genes covering all stages of ILTV transcription, namely *ICP4* (**A**), *ICP27* (**B**), *gI* (**C**), and *gG* (**D**), in LMH cells with p53 overexpression were detected by absolute qRT–PCR at 6 h post ILTV infection. Data are presented as the mean ± SD, *n* = 3. * *p* < 0.05 indicates the levels of significance.

**Figure 3 genes-14-01615-f003:**
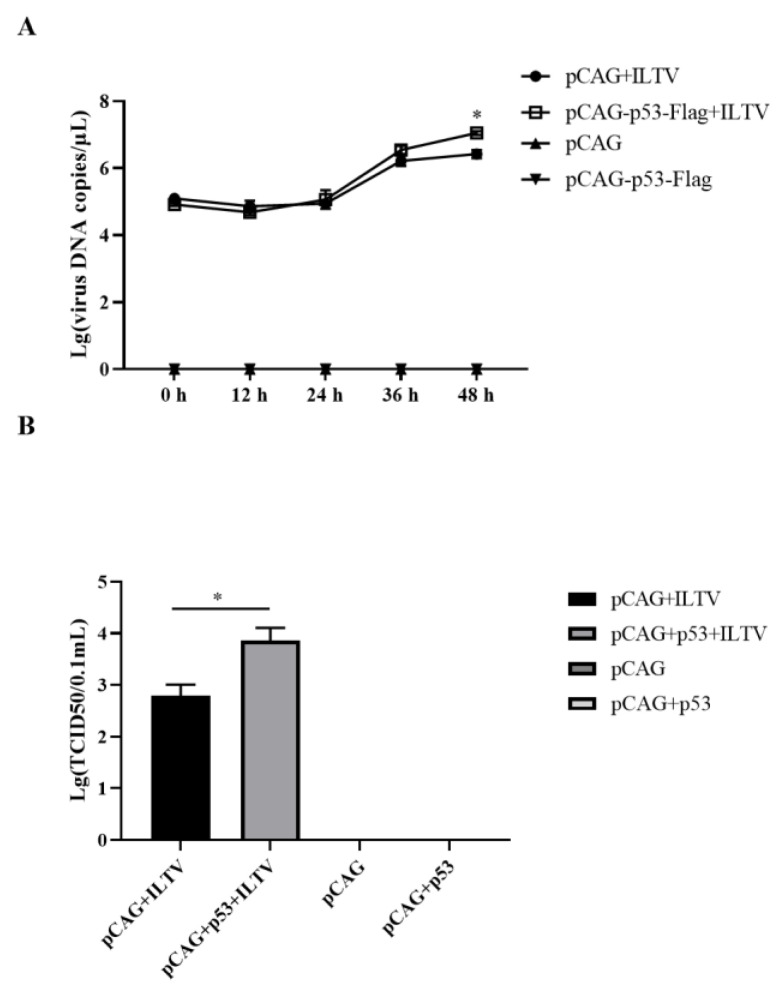
Effects of p53 overexpression on the replication of ILTV in LMH cells. (**A**,**B**) The replication of ILTV in LMH cells was determined by detecting the levels of viral genome replication and infectious virion production using ILTV-specific RT–qPCR assays (**A**) and TCID_50_ assays (48 h post infection) (**B**), respectively. The results are presented as the mean ± SD, *n* = 3. * *p* < 0.05 indicates the levels of significance.

**Figure 4 genes-14-01615-f004:**
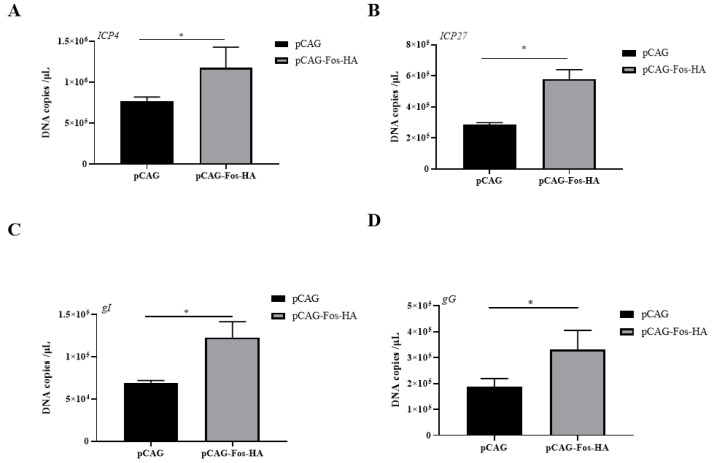
Detection of viral gene transcription upon overexpression of Fos in LMH cells. The mRNA levels of four ILTV genes covering all stages of ILTV transcription, namely *ICP4* (**A**), *ICP27* (**B**), *gI* (**C**), and *gG* (**D**), in LMH cells with Fos overexpression were detected by absolute qRT–PCR at 6 h post ILTV infection. Data are presented as the mean ± SD, *n* = 3. * *p* < 0.05 indicates the levels of significance.

**Figure 5 genes-14-01615-f005:**
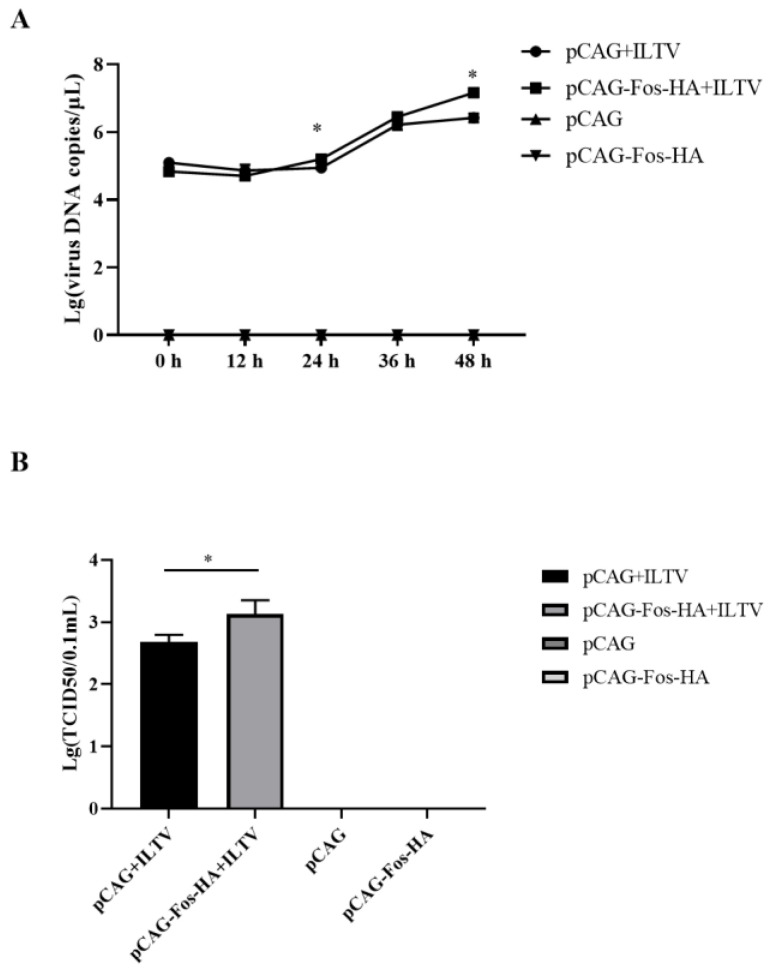
Effects of Fos overexpression on the replication of ILTV in LMH cells. (**A**,**B**) The replication of ILTV in LMH cells was determined by detecting the levels of viral genome replication and infectious virion production using ILTV-specific RT–qPCR assays (**A**) and TCID_50_ assays (48 h post infection) (**B**), respectively. The results are presented as the mean ± SD, *n* = 3. * *p* < 0.05 indicates the levels of significance.

**Figure 6 genes-14-01615-f006:**
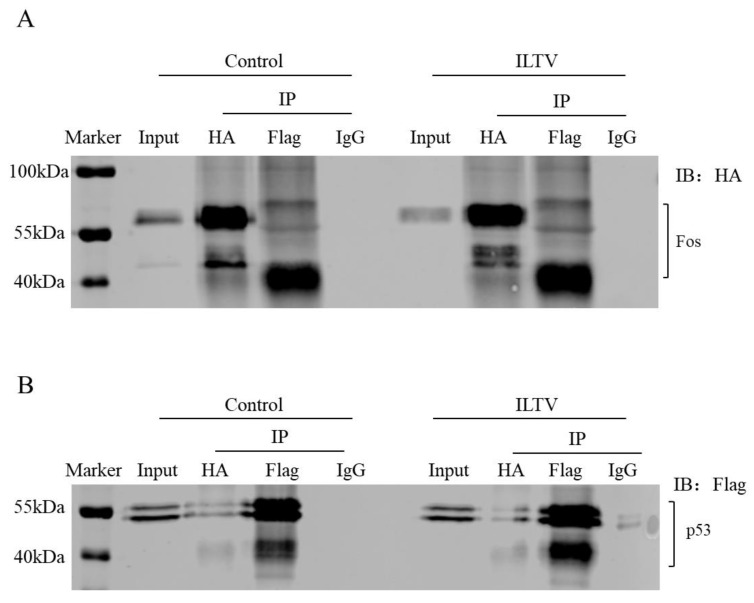
Co-Immunoprecipitation IP experiment revealed direct physical interaction between p53 and Fos proteins in LMH cells. (**A**,**B**) Input and IP samples are detected using HA antibody (**A**) and Flag antibody (**B**), respectively. IP: immunoprecipitation, IB: immunoblotting.

**Figure 7 genes-14-01615-f007:**
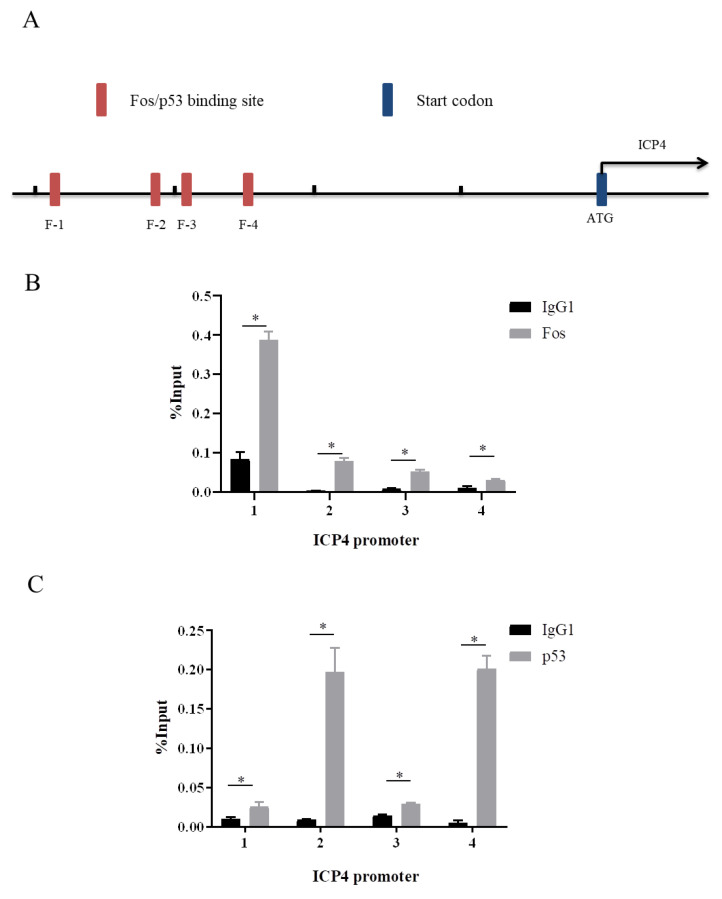
*ICP4* is a bona fide target gene of both p53 and Fos. (**A**) Schematic representation of the putative binding sites of Fos (F-1, F-2, F-3, F-4) in the promoter region of *ICP4*. (**B**,**C**) LMH cells were transfected with a pCAGGS-HA (vector), pCAG-Fos-HA, or pCAGGS-p53-Flag plasmid, respectively, and harvested 12 h after infection with ILTV. ChIP assays were performed with an anti-HA antibody or anti-Flag antibody. DNA input was used as a positive control and IgG1 was used as a negative control. The amount of these four putative binding sites (F-1, F-2, F-3, F-4) bound by p53 and Fos was determined by ChIP-qPCR analysis. Data are presented as the mean ± SD, *n* = 3. * *p* < 0.05 indicates the levels of significance.

**Table 1 genes-14-01615-t001:** Primer sequence for PCR.

Primer Name	Sequence (5′-3′)	Length (bp)
ICP4-F	CGTGGCACTAGATATTAACGTG	503
ICP4-R	CTCGCCAGAGTGGCTCTAGCG	
ICP27-F	CATCTTCGAACTGATGCCAAAGC	441
ICP27-R	CGTCATCACGGACCGAAACGAAGG	
gI-F	CGCCAGGATTGACGACGATCAC	450
gI-R	GTGCGACACGAAGCCTTGGAATAG	
gG-F	ATGAGCGGCTTCAGTAACATAG	511
gG-R	CTGAGAGCTGGTAGGCGTAGATG	
gC-F	CCTTGCGTTTGAATTTTTCTGT	503
gC-R	AATAGCCGGACGACATCTG	

## Data Availability

The data presented in this study are available on request from the corresponding author.

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
