# Peer review of "Characterization of the Effects of Host p53 and Fos on Gallid Alpha Herpesvirus 1 Replication"

_genes, 2023, doi:10.3390/genes14081615_

Round 1

Reviewer 1 Report

General comments

The authors investigated the effects of p53 and Fos on ILTV replication. Both molecules were involved in the expression of viral genes. On the other hand, Fos is not essential for efficient viral replication, whereas p53 contributes to the viral replication. These data will provide a better understanding of the interactions between ILTV and the host. 

However, previously, the authors have already published similar papers (Wang et al., Viruses. 2021; Xu et al., Front in Microbiol. 2022). Novel findings in this manuscript are unclear. In addition, according to a previous study (Wang et al., Viruses. 2021), ILTV replication seemed to be repressed by the knock down of Fos. Why was the difference between them observed?

The authors need to clearly show novel findings different from previous reports.

Other comments

- Do LMH cells have any mutations in the TP53 gene? 

This paper is not a major issue for the quality of the English language.

Author Response

Dear reviewer,

Thank you for your comments and suggestions very much, which are valuable in improving the quality of our manuscript. We adequately considered all comments and suggestions and revised the manuscript accordingly. New data have been added in the revised manuscript according to reviewer’s suggestions. Besides, we proof-read the manuscript to minimize typographical and grammatical errors and omissions. The edited manuscript was the one in the attachment.

Here below is our description on revision according to your comments. All modifications have been highlighted with yellow background in the revised manuscript.

Reviewer’s comments for the authors:

General comments

The reviewer’s comment 1:

The authors investigated the effects of p53 and Fos on ILTV replication. Both molecules were involved in the expression of viral genes. On the other hand, Fos is not essential for efficient viral replication, whereas p53 contributes to the viral replication. These data will provide a better understanding of the interactions between ILTV and the host.

However, previously, the authors have already published similar papers (Wang et al., Viruses. 2021; Xu et al., Front in Microbiol. 2022). Novel findings in this manuscript are unclear. In addition, according to a previous study (Wang et al., Viruses. 2021), ILTV replication seemed to be repressed by the knock down of Fos. Why was the difference between them observed? The authors need to clearly show novel findings different from previous reports.

The authors’ answer:

Thanks for reviewer’s valuable comments and advices! In our previous studies (Wang et al., Viruses. 2021; Xu et al., Front in Microbiol. 2022), the importance of chicken p53 and Fos in ILTV replication have been only validated by knock-down of these two genes. Both other’s and our previous studies showed LMH, the only available chicken cell line permissive for ILTV, as a good model to investigate ILTV infection in vitro. Although LMH cells response well to p53 inhibition, as reviewer mentioned in next comment, the mutation information of p53 in LMH cells remains unclear, which is the same case for Fos. Therefore, it is important to investigate the effects of the overexpression of wild-type p53 and Fos on viral replication in LMH cells, which has been clarified in detail in the present manuscript. We sincerely appreciate reviewer’s valuable indication on Fos knockdown! This inconsistent result was due to the inefficient knockdown of Fos. Different procedure and agent were used in the present study, which leads to unstable knockdown efficiency (as the huge error bar seen in Fig. 1H in previous manuscript). We checked the repeat with efficient Fos knockdown, the same conclusion was obtained with our previous study. In fact, we repeated this experiment with the same protocol used by Dr. Zhitao Wang in our previous study, same results were observed. Considering that both the results of p53 knockdown and Fos knockdown have been reported in our previous studies, these data have been removed from our present manuscript. To strengthen the novelty of current manuscript, new data addressing the potential cooperation between p53 and Fos has been added (Fig. 6 and 7).             

Other comments

The reviewer’s comment 2:

Do LMH cells have any mutations in the TP53 gene?

The authors’ answer:

Thanks for reviewer’s valuable comment! As we mentioned above, it remains unclear whether there is any mutation in the TP53 gene in LMH cells, although LMH cells response well to p53 inhibition during ILTV infection. Therefore, we performed overexpression of wild-type chicken p53 in LMH cells to elucidate the effects of wild-type chicken p53 on ILTV replication in detail.

The reviewer’s comment 3:

This paper is not a major issue for the quality of the English language.

The authors’ answer:

Thanks a lot for reviewer’s patient reading

Reviewer 2 Report

General comments

The authors tried to investigate the impact of chicken p53 and Fos on each 74 stage of ILTV replication by manipulating the expression of p53 and Fos in LMH cells.

 Materials and methods

Line 113 Primer sequences will be made available upon request. I did not understand this sentence.

Line 116 Absolute quantitative real-time PCR. Why did you make absolute quantification? Please explain why did not you make relative expression? You have a control group.

Line 118 Standards for absolute  qRT-PCR were prepared respectively by cloning the PCR products of the ICP4, ICP27, gI,  gG, gC genes of ILTV into the pMD18-T plasmid (Takara, Shiga, Japan) according to the  manufacturer’s instructions. What is the primers sequences for these investigated genes

Line 124: Westernblot. 

Could you emphasize that the checked proteins are species specific? please illustrate.

I have noticed you have made knockdown. Please explain in details in materials and methods section

Discussion

In our previous studies, we identified transcriptional factors p53 and Fos as essential  host determines of ILTV infection. Please add reference. I have noticed this repetition in introduction section. Please add new insights. Avoid repetition.

Discussion section is too short. Provide additional information

Fine English editing is required

Author Response

Dear reviewer,

Thank you for your comments and suggestions very much, which are valuable in improving the quality of our manuscript. We adequately considered all comments and suggestions and revised the manuscript accordingly. We proof-read the manuscript to minimize typographical and grammatical errors and omissions. The edited manuscript was the one in the attachment.

Here below is our description on revision according to your comments. All modifications have been highlighted with yellow background in the revised manuscript.

Reviewer’s comments for the authors:

General comments

The authors tried to investigate the impact of chicken p53 and Fos on each stage of ILTV replication by manipulating the expression of p53 and Fos in LMH cells.

Materials and methods

The reviewer’s comment 1:

Line 113 Primer sequences will be made available upon request. I did not understand this sentence.

The authors’ answer:

Thanks a lot for reviewer’s patient reading and indication! We have removed this sentence, and the specific primers for the genes we detected have been presented in the revised manuscript as Table 1.

The reviewer’s comment 2:

Line 116 Absolute quantitative real-time PCR. Why did you make absolute quantification? Please explain why did not you make relative expression? You have a control group.

The authors’ answer:

Thanks for reviewer’s comment! The absolute quantitative real-time PCR is independent of any inner control and therefore is more accurate than the relative quantitative methods, since viral infection may alter many aspects of host cells, such as the translation, metabolism, and cytoskeleton. Thus, absolute quantitative real-time PCR was used in our manuscript, although there might be no significant difference in the result of detection between these two methods at the beginning of viral infection.

The reviewer’s comment 3:

Line 118 Standards for absolute  qRT-PCR were prepared respectively by cloning the PCR products of the ICP4, ICP27, gI,  gG, gC genes of ILTV into the pMD18-T plasmid (Takara, Shiga, Japan) according to the  manufacturer’s instructions. What is the primers sequences for these investigated genes

The authors’ answer:

Thanks a lot for reviewer’s patient reading and indication! The primer sequences have been provided in the revised manuscript as Table 1.

The reviewer’s comment 4:

Line 124: Westernblot. Could you emphasize that the checked proteins are species specific? please illustrate. I have noticed you have made knockdown. Please explain in details in materials and methods section

The authors’ answer:

Thanks for reviewer’s questions! The purpose of Westernblot detection in our manuscript is to evaluate the efficiencies of chicken p53 overexpression and Fos overexpression. For the overexpression of chicken p53 or Fos, LMH cells were transiently transfected with pCAG-Fos-HA or pCAG-p53-Flag. The protein levels of overexpressed p53 and Fos were detected by antibody specifically recognizing HA or Flag. Because the results of knockdown of p53 and Fos have been reported in our previous studies, this part of the study has been removed from the revised manuscript. To strengthen the novelty of current manuscript, new data addressing the potential cooperation between p53 and Fos has been added (Fig. 6 and 7).

Discussion

The reviewer’s comment 5:

In our previous studies, we identified transcriptional factors p53 and Fos as essential host determines of ILTV infection. Please add reference. I have noticed this repetition in introduction section. Please add new insights. Avoid repetition.

The authors’ answer:

Thanks for reviewer’s valuable comment! In the revised manuscript, two related references were cited here. The sentence with high repetition with introduction section has been modified according to reviewer’s suggestion.

References cited here:

  1. Zhitao Wang, et al. Fos Facilitates Gallid Alpha-Herpesvirus 1 Infection by Transcriptional Control of Host Metabolic Genes and Viral Immediate Early Gene. Viruses. 2021 Jun 9;13(6):1110.
  2. Xu Li, et al. P53 maintains gallid alpha herpesvirus 1 replication by direct regulation of nucleotide metabolism and ATP synthesis through its target genes. Frontiers in Microbiology, 2022(13), 1044141.

The reviewer’s comment 6:

Discussion section is too short. Provide additional information

The authors’ answer:

Thanks for reviewer’s kind advice! New contents have been added to the discussion section of the revised manuscript. All new additions have been highlighted.

The reviewer’s comment 7:

Fine English editing is required

The authors’ answer:

Thanks a lot for reviewer’s patient reading! The manuscript has been polished by our colleague who is native English speaker to check the English. Besides, we proof-read the manuscript to minimize typographical and grammatical errors and omissions.

Round 2

Reviewer 1 Report

The authors added new data to show the novelty of the paper.

In addition, the answers to the points raised by the referee clarified the reasons for the discrepancies with the data in previous papers.

Based on the above, I consider that the authors have provided answers to the referee's questions.

However, a question remains regarding the additional data.

The authors must clearly clarify the following points.

For Figure 6, each molecule should be indicated by arrows.

The molecular sizes of each molecule should also be indicated.

The anti-HA antibody can detect HA-tagged Fos protein, whereas anti-FLAG antibodies should detect FLAG-tagged p53 proteins.

In figure 6A, the authors detected HA-tagged Fos proteins using anti-HA antibodies.

Thus, if HA-tagged Fos was co-precipitated with FLAG-tagged p53, HA-tagged Fos should be detected in the IP-FLAG lane by the anti-HA antibody.

However, it is not clear which bands indicate HA-tagged Fos in the lanes of IP-Flag.

Author Response

Dear reviewer,

Thank you for your comments and suggestions very much, which are helpful for revising and improving our paper! We adequately considered the comment and revised the manuscript accordingly. The edited manuscript was the one in the attachment.

Here below is our description on revision according to the comment.

Reviewer’s comments for the authors:

The authors added new data to show the novelty of the paper.In addition, the answers to the points raised by the referee clarified the reasons for the discrepancies with the data in previous papers. Based on the above, I consider that the authors have provided answers to the referee's questions.

The reviewer’s comment 1:

However, a question remains regarding the additional data. The authors must clearly clarify the following points. For Figure 6, each molecule should be indicated by arrows. The molecular sizes of each molecule should also be indicated. The anti-HA antibody can detect HA-tagged Fos protein, whereas anti-FLAG antibodies should detect FLAG-tagged p53 proteins. In figure 6A, the authors detected HA-tagged Fos proteins using anti-HA antibodies. Thus, if HA-tagged Fos was co-precipitated with FLAG-tagged p53, HA-tagged Fos should be detected in the IP-FLAG lane by the anti-HA antibody. However, it is not clear which bands indicate HA-tagged Fos in the lanes of IP-Flag.

The authors’ answer:

Thanks for reviewer’s comment! In the newly revised manuscript, we have mapped in detail which bands represent HA tagged Fos and Flag tagged p53 and the molecule sizes were indicated in the results. This HA-tag and FLAG-tag system is widely used in our lab, since the antibodies targeting HA or FLAG show good specificity in our models. The predicted size of our chicken Fos-HA is about 41kDa. By western blot, multiple binds of overexpressed Fos ranging from 40 kDa to 60 kDa were observed. The same thing happens for human Fos, since many widely used monoclonal antibodies also detect multiple binds on gels, such as clone N486/32 (ab302667) from Abcam, clone E2I7R (#31254) from CST, and clone K25 (sc-253) from Santa cruze. The multiple binds of Fos may due to either the post-translational modification (phosphorylation, glycosylation etc) which increases the size of Fos or post-translation cleavage and splice variants which may create different sized proteins presented as lower binds [1]. The size of Fos on gels may various among different cell types and cellular conditions. In our results, it is interesting that p53 mainly binds to the low bind of Fos. The same with that of Fos, multiple binds were observed for p53, although the predicted size of p53-Flag is about 43kDa. The high binds may due to the post-translational modifications such as phosphorylation, acetylation, and methylation, which regulate p53 activity, localization, stability, and functions [2].       

Reference:

  1. Yu, C. T., Wu, J. C., Liao, M. C., Hsu, S. L., & Huang, C. Y. (2008). Identification of c-Fos as a mitotic phosphoprotein: regulation of c-Fos by Aurora-A. Journal of biomedical science, 15(1), 79–87.
  2. Liu, Y., Tavana, O., & Gu, W. (2019). p53 modifications: exquisite decorations of the powerful guardian. Journal of molecular cell biology, 11(7), 564–577.

Examples of the multiple binds of human Fos detected by commercial monoclonal antibodies:

Reviewer 2 Report

The authors have made my suggestions so i accept the manuscript in its current form

Author Response

Dear reviewer,

Thank you for your comments very much! We appreciate the support you have provided, which improved our paper a lot.

Reviewer’s comments for the authors:

The reviewer’s comment 1:

The authors have made my suggestions so I accept the manuscript in its current form

The authors’ answer:

We thank the reviewer again for the valuable suggestions and for the appreciation of our work!

Round 3

Reviewer 1 Report

The authors have addressed my concerns. I agree that this manuscript should be accepted for the publication.

fine